# Decreased Physical and Daily Living Activities in Patients with Peripheral Arterial Disease on Hemodialysis

**DOI:** 10.3390/jcm12010135

**Published:** 2022-12-24

**Authors:** Yuma Tamura, Harunori Takahashi, Daiki Sakai, Tomoki Tsurumi, Hajime Tamiya, Asuka Ueno, Shinya Kawamoto, Masahiro Shimoyama, Takanori Yasu

**Affiliations:** 1Department of Rehabilitation, Dokkyo Medical University Nikko Medical Center, Nikko 321-2593, Japan; 2Department of Physical Therapy, Niigata University of Health and Welfare, Niigata 950-3198, Japan; 3Department of Cardiovascular Medicine and Nephrology, Dokkyo Medical University Nikko Medical Center, Nikko 321-2593, Japan

**Keywords:** hemodialysis, peripheral arterial disease, life-space assessment, difficulty in activities of daily life

## Abstract

Decreased physical activity and activities of daily living (ADL) in patients on hemodialysis (HD) are associated with a poor prognosis. Additionally, comorbid peripheral arterial disease is associated with further deterioration. We conducted a cross-sectional study of ADL difficulty and life-space assessment (LSA) in three groups of patients on hemodialysis according to their ankle-brachial index (ABI) values. The 164 patients were divided into ABI Low (ABI < 0.9), Normal (0.9 ≤ ABI < 1.3), and High (1.3 ≤ ABI) groups, and compared using analysis of covariance with LSA and ADL difficulty adjusted for age. The Kihon checklist (KCL) was used to assess the presence of frailty. The LSA was lower in the Low group than in the High group (F = 3.192, *p* = 0.044). Similarly, the ADL difficulty was significantly lower in the Low group than in the Normal group (F = 3.659, *p* = 0.028). In the Low group, the proportion of patients with frailty was 47.1% and KCL physical was significantly lower, indicating that patients on HD with a lower ABI had a higher prevalence of frailty and lower LSA and ADL difficulty. In conclusion, patients on HD with decreased ABI values most likely exhibit decline in physical activity and ADL due to frailty and claudication.

## 1. Introduction

Patients on hemodialysis (HD) have a higher incidence of cardiovascular disease and stroke, as well as a higher rate of peripheral arterial disease (PAD), an atherosclerotic disease, than those not on dialysis [1]. Approximately 16–19% of Japanese patients on HD have an ankle-brachial index (ABI) of less than 0.9 [2,3], which is a criterion for PAD, and 4.7% have PAD in the lower extremities at the time of HD induction [4]. Diabetic kidney disease (DKD) is the most common inductive cause for dialysis in Japan. Moreover, arteriosclerosis and microcirculatory disturbances caused by diabetes mellitus (DM) also play a major role in the development of cardiovascular disease [5,6].

HD and PAD both trigger reduced physical activity [7] and are associated with reduced survival [8]. The pain caused by PAD and the fatigue caused by HD may amplify physical inactivity. Life-space assessment (LSA), which measures the extent of living space, has been reported to decrease with progressive deterioration of renal function [9], and is an indicator of physical activity, suggesting a decrease in patients on HD. Mortality is significantly higher in patients with difficulty in mobility tasks, such as walking while performing activities of daily living (ADL) [10], and gait disturbance due to PAD is a factor contributing to worsening their ADL. In other words, ADL difficulty and LSA are expected to deteriorate further in patients on HD with PAD; however, the details are unclear. The proportion of frail patients on HD with PAD has significantly increased [11], and since frailty is a primary component of physical inactivity [12], further reduction in physical activity and ADL through frailty remains a concern. In addition, DKD is an important risk factor for frailty, and the prevalence of sarcopenia, a component of physical frailty, increases with worsening DKD [13].

This study aimed to evaluate PAD with ABI values in patients on HD and compare their ADL difficulty and LSA to determine whether there is worsening due to PAD complications.

## 2. Materials and Methods

### 2.1. Study Population

The participants were patients on chronic HD who regularly visited the Yuai clinic during the daytime within the period of this study from February 2020 to March 2021. The initial exclusion criteria were as follows: (1) refusal to participate, (2) hospitalization due to accidents or sickness, and (3) difficulty in answering the questionnaire. Participants who did not complete the questionnaire were also excluded from the study. Figure 1 shows the selection of participants in this study. Of the 231 patients on HD, 50 were excluded due to refusal to participate in the study, hospitalization, or difficulty in answering the questionnaire, and another 17 were excluded due to inadequate responses to the questionnaire. A total of 164 patients were enrolled in the study. They were divided into three groups according to ABI values: Low (<0.9), Normal (≥0.9, <1.3), and High (≥1.3).

This study was performed in accordance with the principles of the Declaration of Helsinki and approved by the Institutional Ethics Committee of Dokkyo Medical University (approval number: Nikko 20-024). Written informed consent was obtained from the patients prior to participation in the study.

### 2.2. Evaluation Measurements and Factors

The following items were calculated from the medical records as clinical characteristics of the patients: body mass index (BMI), smoking habits, number of oral medicines, frequency and quantity of dialysis, calculated single-pooled Kt/V [14], and blood test data. Preexisting medical conditions were investigated for DM, hypertension, dyslipidemia, myocardial infarction, and stroke, with reference to previous reports [15]. ABI and brachial-ankle plus wave velocity results were examined bilaterally in the lower extremities. Nutritional status was assessed using the Geriatric Nutritional Risk Index (GNRI).

#### 2.2.1. ADL Difficulty

ADL difficulty was measured using a previously reported questionnaire developed for patients on HD [10,16]. This questionnaire assessed difficulties in ADL mobility and consisted of the following 12 items: “rising from a chair,” “rising from the floor,” “sitting down on the floor,” “walking 100 m,” “walking 300 m,” “walking 600 m,” “walking 1 km,” “walking 20 m quickly,” “walking up one flight of stairs,” “walking up two flights of stairs,” “walking down one flight of stairs,” and “walking down two flights of stairs.” The 12 items were divided into the following three categories: “basic ADL,” “ambulation,” and “walking up or down.” Patients responded to these items with 1–5 points of perceived difficulty (1, not possible; 2, severe difficulty; 3, moderate difficulty; 4, mild difficulty; and 5, ease). The ADL score was calculated from the sum of the 12 items, with a maximum score of 60 points. Lower scores indicate lower ADL.

#### 2.2.2. Life-Space Assessment

LSA is an indicator of mobility and social participation in an individual’s life space expanse [17,18]. The sub-score for life-space mobility consisted of five items: (1) home, (2) outside the house, (3) neighborhood, (4) town, and (5) outside the town. The respondents were asked about the frequency of mobility and use of assistive devices. The specific questions to be asked were as follows: (1) “In the past 4 weeks, did you go to any room in your home other than the room where you sleep (Level 1)?,” (2) “In the past 4 weeks, did you go outside your front door, balcony, apartment hallway, garage, yard, or walkway on your property (Level 2)?,” (3) “In the past 4 weeks, did you go to any neighborhood area outside your yard or apartment building (Level 3),?” (4) “In the past 4 weeks, did you go to places further away than your neighborhood (Level 4)?,” and (5) “In the past 4 weeks, did you go outside of town (Level 5)?” For each life space level, subjects were asked how often they traveled to the area (less than once a week, 1–3 times a week, or 4–6 times a week, daily) and whether they needed help from other people or assistive devices (“yes” vs. “no”) The total LSA score was calculated by scoring each activity area, with activities inside the house scoring the lowest. The total score of the LSA was calculated by scoring each activity in each range, with a maximum score of 8 points for activities in the home, 16 points for activities around the home, 24 points for activities in the neighborhood, 32 points for activities in the town, and 40 points for activities outside the town. The scores ranged from 0 to 120 points.

#### 2.2.3. Kihon Checklist (KCL)

The KCL is a questionnaire-based assessment method designed to screen high-risk older adults who are likely to require long-term care [19]. It consists of 25 questions about physical function and mental status and has seven domains: 1–5, ADL; 6–10, physical activities and falls; 11–12, nutrition, 13–15, oral functions, 16–17, outdoor activities, 18–20, cognitive function, and 21–25, depression. All questions were answered on a Yes-or-No scale, with a score of 25 points: 0–3 points were considered “robust,” 4–7 points were considered “pre-frailty,” and ≥8 points were considered “frailty” [20,21]. When restricted to physical frailty, a cut-off value of >3 points was used in items 6–10 [20,22]. In addition, items 1–20 were defined as physical and items 21–25 as depression, and comparisons were made between groups.

### 2.3. Statistical Analysis

Data for continuous variables are expressed as the mean ± standard deviation based on the normality of the distribution assessed using the Shapiro–Wilk test. Categorical variables are expressed as numbers (*n*) and percentages, and χ^2^ tests were used for comparisons by group. Differences among phenotypes were examined using one-way analysis of variance or the Kruskal–Wallis test. Age-adjusted analysis of covariance was performed to evaluate the independent association of each ABI group. When one-way analysis of variance showed a statistically significant difference, a post hoc test was performed using the Bonferroni test. LSA and ADL difficulty were examined for associated factors using Pearson’s correlation analysis. Multiple regression analysis was performed using a stepwise method, with the correlated items being considered as independent variables in consideration of multicollinearity. For all tests, statistical significance was set at *p* < 0.05. All analyses were performed using the SPSS Statistics software (IBM Corp., Armonk, NY, USA).

## 3. Results

### 3.1. Demographic and Clinical Data

The clinical characteristics of 164 patients are presented in Table 1. The patients’ mean age, BMI, and HD duration were 62.7 ± 11.2 years, 22.5 ± 4.3 kg/m^2^, 157.3 ± 120.7 months, respectively with 72.0% men and 28.0% women. The patients were divided into three groups according to their ABI values: Low (ABI < 0.9, *n* = 17, 10.3%), Normal (0.9 ≤ ABI < 1.3, *n* = 97, 59.1%), and High (ABI ≥ 1.3, *n* = 50, 30.5%). In individuals with ABI ≥ 1.3, vascular calcification is advanced and ABI alone is not sufficient to evaluate PAD. Therefore, the toe-brachial index or skin perfusion pressure was measured in all patients with ABI ≥1.3 to confirm that they were in the normal range and did not have PAD.

The three groups showed significant differences in terms of sex, BMI, dry weight, water removal, creatinine, and DM comorbidity. However, no significant differences were observed in terms of age.

### 3.2. Life-Space Assessment (LSA)

The mean LSA was 73.1 ± 23.0 for all patients, with 58.4 ± 23.2 for the Low, 73.7 ± 21.8 for the Normal, and 74 ± 23.6 for the High group. When the groups were adjusted for age, there was a significant decrease in LSA in the Low group compared to that in the High group (Figure 2). Correlation items for LSA were shown in Appendix A. Multiple regression analysis showed that the factors that were significantly correlated with LSA were dialysis duration (β = −0.216, *p* = 0.001), GNRI (β = 0.156, *p* = 0.024), and KCL total (β = −0.452, *p* < 0.001) (R^2^ = 0.360) (Table 2).

### 3.3. ADL Difficulty

The mean total score of ADL difficulty was 48.2 ± 11.9 for all patients, with 39.9 ± 14.7 for the Low, 49.5 ± 11.0 for the Normal, and 48.5 ± 11.8 for the High group, showing a significant decrease in the Low compared to those in the Normal group (Figure 3). The scores for these three subcategories are presented in Table 3. Walking up or down was significantly worse in the Low group than that in the Normal and High groups. The percentages of the responses for the three groups are shown in the upper panel of Figure 4. Correlation items for ADL difficulty were shown in Appendix A. Multiple regression analysis showed that the factors that were significantly correlated with ADL difficulty were age (β = −0.213, *p* < 0.001) and KCL total (β = −0.669, *p* < 0.001) (R^2^ = 0.590) (Table 2).

### 3.4. Nutritional Index

The GNRI scores were not significantly different among the three groups (Table 3). The percentage of those with no nutritional disability (GNRI ≥ 98) was 2.4 % for the Low, 53.7% for the Normal, and 43.9% for the High group and tended to be lower in the Low group (Figure 4, middle panel).

### 3.5. Frailty

There were no significant differences in the total KCL scores; however, significant differences were found in the KCL physical scores in the sub-items (Table 3). The percentage of frailty in each group is shown in the lower part of Figure 4. The prevalence of frailty was 47.1% for the Low, 30.8% for the Normal, and 32.0% for the High group.

## 4. Discussion

To our best knowledge, this is the first report to show that patients on HD have low physical activity and increased ADL difficulty, which is further exacerbated by the presence of PAD.

Patients on HD have a higher prevalence of PAD [1] and patients with PAD at the start of HD have a poorer prognosis than those without PAD [4]. The prognosis of patients on HD with a physical activity level of less than 4000 steps/day is significantly poor [8]. PAD with intermittent claudication is also a disorder that leads to decreased physical activity [7], and the concomitant occurrence of PAD in patients on HD is thought to synergistically cause difficulties in physical activity and movement.

In this study, patients on HD may have a high ABI due to vascular calcification and were divided into three groups according to ABI values in previous reports. LSA is also used as a measure of physical activity that worsens as nephropathy progresses [23].Previous reports on patients on HD have shown a mean LSA of 79 ± 26, similar to that in our study patients [9]. In this study, low ABI decreased LSA, suggesting that physical activity decreases with the frequency of outings and narrowing of the range of outings. The amount of physical activity in patients on HD declines with age [24], and the LSA may also be affected by age. However, these patients showed a decrease in LSA in the Low group, even when adjusted for age, and we believe that the presence or absence of PAD affects the decrease in LSA independently of aging factors.

Similarly, ADL difficulty also declined in the Low group, even when adjusted for age, and lower ADL difficulty in patients on HD is a poor prognostic factor [10]. ADL disfunction is associated with an increased risk of mortality in patients on HD [25]. ADL difficulty evaluates the state in which a patient is able to perform an operation, but with increasing difficulty as a preliminary step to the occurrence of ADL decline. In other words, it may be possible to detect ADL decline at an earlier stage [10]. The ADL difficulty subscale was also compared in this study; the symptoms of PAD were accompanied by intermittent claudication, and the difficulty in performing ADL activities increased because of the reduced gait function. Therefore, the ADL difficulty sub-items showed a significant increase in difficulty in activities involving walking, such as “basic-ADL” and “walking up or down.” In the walking-impairment questionnaire, an association was found between staircase items and walking speed, indicating that gait disturbance due to PAD increases ADL difficulty [26].

Thus, patients on HD have comorbid PAD, which leads to decreased LSA and increased ADL difficulty, which may be due to progressive frailty. Multiple regression analysis showed that both LSA and ADL difficulty were associated with KCL total, confirming the effect of frailty. Patients on HD with frailty had a 2.06-fold higher risk of developing PAD, and 34% of patients on HD with comorbid PAD had frailty [11]. The complication rate of frailty in the Low group in this study was 52.9%, which is higher than that previously reported. In addition, patients on HD with PAD have decreased leg muscle strength and grip strength [11], and further deterioration of frailty is thought to lead to a decrease in LSA and increase in ADL difficulty. In this study, the Low group had a higher incidence of DM, a factor strongly associated with both frailty and PAD development. This suggested that DKD-derived patients on HD had worse physical function. Furthermore, malnutrition in patients on HD is associated with the development of atherosclerotic diseases. In other words, factors such as protein restriction in chronic kidney disease and amino acid leakage due to HD have been reported to cause malnutrition-influenced atherosclerosis syndrome and contribute to the development of atherosclerotic disease [27]. In this study, there were many cases of nutritional disorders in the Low group, which may be related to the progression of PAD. GNRI was also extracted as an associated factor for LSA, suggesting that malnutrition is associated with worse physical activity. Rehabilitation was useful in improving ADL and peak oxygen consumption in patients on HD [28,29]. Patients on HD who had weak lower limb muscles, limited mobility, and a 50% decrease in the functional independence measure score (an ADL index) also showed improvement in motor items, especially when physical therapists made them practice standing and walking and occupational therapists made them practice dressing and toileting [29]. In other words, rehabilitation may be successful for fatigue due to dialysis and mobility impairments due to PAD. At the same time, prevention of frailty, nutritional therapy, and control of PAD progression may be useful for improving prognosis. The present study had some limitations. First, the number of patients, especially the Low group was limited. Second, physical activity was assessed using a questionnaire, and the actual physical activity level was unknown. Therefore, it is necessary to evaluate the level of improvement in gait function with PAD rehabilitation in more detail, as well as the level of improvement in the amount of physical activity and ADL, and to examine these relationships while considering the underlying diseases as well.

## 5. Conclusions

Patients on HD with decreased ABI showed more decreased LSA and increased ADL difficulty and were more likely to have frailty. We suggest that gait disturbance due to PAD is responsible for the decline in physical activity and ADL.

## Figures and Tables

**Figure 1 jcm-12-00135-f001:**
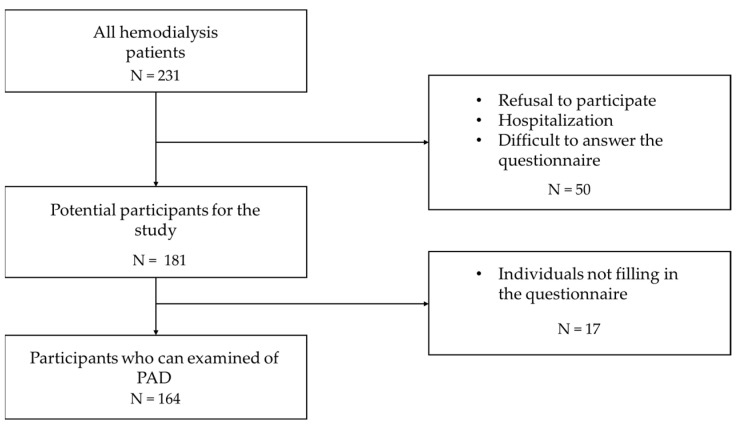
Flowchart of participant selection for the analysis. Of the 231 hemodialysis patients, 67 were excluded from the study and 164 participated. PAD, peripheral arterial disease.

**Figure 2 jcm-12-00135-f002:**
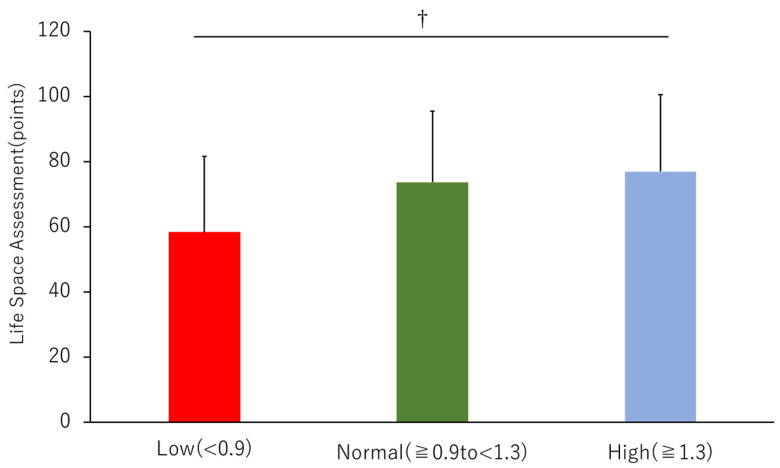
Comparison of life-space assessment by ankle-brachial index (ABI). Red bars indicate an ABI < 0.9, green bars indicate an ABI ≥ 0.9, but <1.3, and blue bars indicate an ABI ≥ 1.3, adjusted for age and patient characteristics to show statistically significant differences in analysis of covariance (ANCOVA). †, vs. High group; *p* < 0.05.

**Figure 3 jcm-12-00135-f003:**
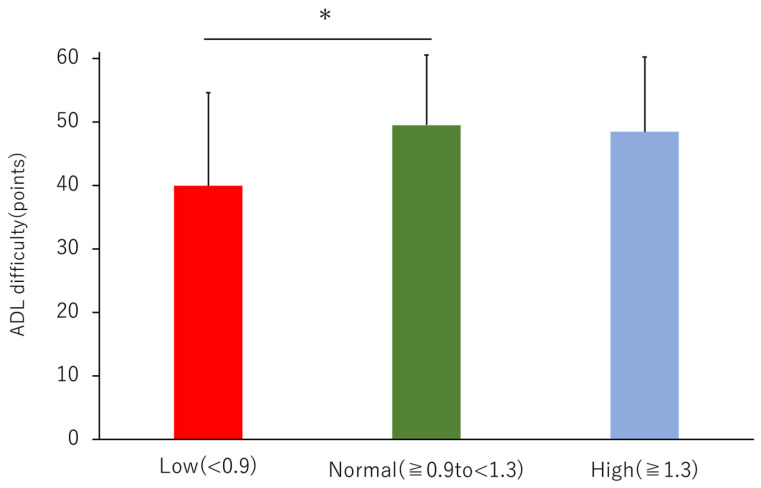
Comparison of activity of daily living-difficulty by ABI. Red bars indicate an ABI < 0.9, green bars indicate an ABI ≥ 0.9, but <1.3, and blue bars indicate an ABI ≥ 1.3, adjusted for age and patient characteristics to show statistically significant differences in ANCOVA. *, vs. Normal group; *p* < 0.05.

**Figure 4 jcm-12-00135-f004:**
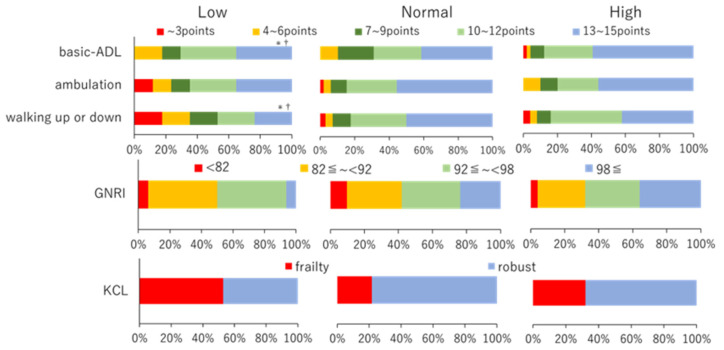
Percentage of ADL difficulty, Geriatric Nutritional Risk Index (GNRI), and Kihon check list (KCL) by classification. The basic ADL is shown in the upper panel. Red, 3 points; orange, 4–6 points; green, 7–9 points; yellow/green, 10–12 points; blue, 13–15 points. GNRI is depicted in the middle panel. Red, severe risk (<82); orange, moderate risk (82–92); yellow/green, mild risk (92–98); blue, without risk (≥98). The KCL is represented in the lower panel. Red, frail; blue, robust. *, vs. the Normal group, *p* < 0.05; ^†^, vs. the High group, *p* < 0.05.

**Table 1 jcm-12-00135-t001:** Patients’ clinical characteristics.

Characteristics	All (*n* = 164)	Low (*n* = 17)	Normal (*n* = 97)	High (*n* = 50)	*p*
Age	62.7 ± 11.2	67.9 ± 12.0	61.8 ± 11.7	62.6 ± 9.5	0.113
Male, *n* (%)	118 (72.0%)	9 (52.9%) ^†^	66 (68.0%)	43 (86.0%)	0.013
BMI (kg/m^2^)	22.5 ± 4.3	21.1 ± 3.3 ^†^	22.3 ± 4.5 †	22.5 ± 4	0.090
Smoking history, *n* (%)	30 (18.2%)	4 (23.5%)	15 (15.5%)	11 (22.0%)	0.526
Number of oral medicines	8.1 ± 2.9	9.6 ± 3.8	7.8 ± 2.7	8.1 ± 2.8	0.067
Dialysis sessions/week	3.0 ± 0.1	2.9 ± 0.2	3.0 ± 0.1	3.0 ± 0	0.288
Dialysis duration (month)	157.3 ± 120.7	171.7 ± 134.8	150.9 ± 107.7	164.8 ± 139.9	0.703
Dialysis time (hour)	4.0 ± 0.3	3.9 ± 0.3	4.0 ± 0.3	4.0 ± 0.3	0.645
Kt/V sp	1.5 ± 0.2	1.5 ± 0.2	1.5 ± 0.3	1.4 ± 0.2	0.092
Dry weight	61.6 ± 15.4	55.2 ± 11.6 ^†^	59.9 ± 15.3†	67.1 ± 15.4	0.005
Water removal	2.8 ± 1.1	2.1 ± 0.6 *^,†^	2.8 ± 1.1	2.9 ± 1.2	0.022
Hb (g/dL)	11.0 ± 1.1	10.7 ± 0.9	11.1 ± 1.2	11.2 ± 1.1	0.277
Alb (g/dL)	3.6 ± 0.4	3.4 ± 0.3	3.6 ± 0.4	3.7 ± 0.3	0.132
BUN (mg/dL)	65.3 ± 14.1	58.5 ± 16.0	66.3 ± 14.0	65.5 ± 13.3	0.105
Cre (mg/dL)	11.7 ± 2.9	9.8 ± 1.9 *†	11.9 ± 3.1	12.1 ± 2.6	0.013
HDL-c	44.3 ± 14.4	40.9 ± 11.7	45.2 ± 15.2	44.0 ± 14.0	0.512
LDL-c	85.8 ± 28.5	75.4 ± 25.4	89.1 ± 31.0	82.8 ± 23.4	0.128
DM, *n* (%)	68 (41.4%)	12 (70.6%) *	36 (37.1%)	20 (40.0%)	0.035
HTN, *n* (%)	97 (59.1%)	8 (47.1%)	60 (61.9%)	29 (58.0%)	0.511
DLP, *n* (%)	13 (7.9%)	2 (11.8%)	5 (5.2%)	6 (12.0%)	0.289
MI, *n* (%)	19 (11.6%)	3 (17.6%)	10 (10.3%)	6 (12.0%)	0.681
stroke, *n* (%)	28 (17.1%)	2 (11.8%)	19 (19.6%)	7 (14.0%)	0.577
ABI right	1.2 ± 0.2	0.9 ± 0.2	1.1 ± 0.1	1.3 ± 0.1	<0.001
ABI left	1.2 ± 0.2	0.9 ± 0.2	1.1 ± 0.1	1.3 ± 0.1	<0.001
ba PWV right (cm/s)	1761 ± 467.5	1978.1 ± 679.6	1731.7 ± 434.5	1755.8 ± 447.2	0.164
ba PWV left (cm/s)	1790.6 ± 543.7	2017.4 ± 755.2	1756.3 ± 532.3	1784.8 ± 477.9	0.205

BMI, body mass index; Kt/V, K reflects clearance of BUN, t reflects dialysis time, and V reflects body fluid volume; Hb, hemoglobin; Alb, albumin; BUN, blood urea nitrogen; Cre, creatinine; HDL-c, high density lipoprotein cholesterol; LDL-c, low density lipoprotein cholesterol; DM, diabetes mellitus; HTN, hypertension; DLP, dyslipidemia; MI, myocardial infarction; ABI, ankle-brachial pressure index; ba PWV, brachial-ankle pulse wave velocity. *p* values were obtained using one-way analysis of variance and Kruskal–Wallis test; *p* *: vs. Normal group with *p* < 0.05. *p* ^†^: vs. High group with *p* < 0.05, as tested by a Bonferroni or a Mann–Whitney U test.

**Table 2 jcm-12-00135-t002:** Multiple regression analysis for LSA and ADL difficulty.

LSA				ADL Difficulty			
Factor	Standard Coefficient β	Standard Error	*p*	Factor	Standard Coefficient β	Standard Error	*p*
Dialysis duration	−0.216	0.013	0.001	Age	−0.213	0.059	<0.001
GNRI	0.156	0.171	0.024	KCL total	−0.669	0.139	<0.001
KCL total	−0.452	0.332	<0.001				
Adjusted R^2^	0.360			Adjusted R^2^	0.590		

The dependent variable was ADL difficulty or LSA and the independent variables were age, BMI, water removal, dialysis duration, GNRI, and KCL total. LSA, life-space assessment; GNRI, Geriatric Nutritional Risk Index; KCL, Kihon check list; BMI, body mass index.

**Table 3 jcm-12-00135-t003:** Patients’ GNRI, ADL difficulty, and KCL outcomes.

Characteristics	All (*n* = 164)	Low (*n* = 17)	Normal (*n* = 97)	High (*n* = 50)	*F*	*p*
GNRI	92 ± 9.2	89.4 ± 6.7	91.2 ± 10.5	94.5 ± 6.4	2.914	0.057
ADL difficulty-basic ADL (points)	12.8 ± 2.9	11.1 ± 3.5 *	13.1 ± 2.8	12.7 ± 2.8	3.659	0.028
ADL difficulty-ambulation (points)	19.9 ± 5.5	16.9 ± 6.8	20.4 ± 5.1	20.0 ± 5.6	2.954	0.055
ADL difficulty-walking up or down (points)	15.5 ± 4.4	12.0 ± 5.7 *^,†^	16.1 ± 3.9	15.7 ± 4.2	6.747	0.002
KCL total (points)	6.4 ± 4.7	8.6 ± 6.0	5.9 ± 4.5	6.6 ± 4.4	2.953	0.090
KCL physical (points)	4.9 ± 3.6	6.8 ± 4.3 *	4.5 ± 3.5	5.0 ± 3.4	3.823	0.048
KCL depression (points)	1.5 ± 1.6	1.8 ± 2.0	1.4 ± 1.5	1.6 ± 1.8	0.518	0.596

GNRI, Geriatric Nutritional Risk Index; KCL, Kihon check list. *p* values are obtained by one way analysis of variance test or Kruskal–Wallis test. *p* *: vs. Normal group with *p* < 0.05. *p* ^†^: vs. High group with *p* < 0.05 by Bonferroni test.

## Data Availability

Not applicable.

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
