# Peer review of "Decreased Physical and Daily Living Activities in Patients with Peripheral Arterial Disease on Hemodialysis"

_jcm, 2022, doi:10.3390/jcm12010135_

Round 1
Reviewer 1 Report
This is an intreesting and well written paper. The one and only issue I see is the following:
Authors conducted several descriptive analyses, using different variables. Partly three investiagated groups are different in terms of baseline characteristics. With these data authors have a perfect chance to perform multivariable regression analysis, not only age-adjusted descriptive test. What is the reason why authors decided to avoid these regression analyses? These analyses would strongly strengthen paper.
Reviewer 2 Report
dear authors thank you for the submission. the paper is very well written.
1)clarify the inclusion and exclusion criteria
2)in the discussion explain and expand that it may be possible to develop targeted management and/or treatment approaches by addressing the immediate symptoms/conditions that individuals attribute their difficulties with ADLs to. maybe by physical and occupational therapists.
3)i suggest to publish the dataset and analysis codes as supplumental
Round 2
Reviewer 1 Report
I thank authors for new analyses